# The Effects of Speed and Delays on Test-Time Performance of End-to-End Self-Driving

**DOI:** 10.3390/s24061963

**Published:** 2024-03-19

**Authors:** Ardi Tampuu, Kristjan Roosild, Ilmar Uduste

**Affiliations:** Insititute of Computer Science, University of Tartu, 51009 Tartu, Estonia

**Keywords:** end-to-end driving, delays, out of distribution, ADS testing

## Abstract

This study investigates the effects of speed variations and computational delays on the performance of end-to-end autonomous driving systems (ADS). Utilizing 1:10 scale mini-cars with limited computational resources, we demonstrate that different driving speeds significantly alter the task of the driving model, challenging the generalization capabilities of systems trained at a singular speed profile. Our findings reveal that models trained to drive at high speeds struggle with slower speeds and vice versa. Consequently, testing an ADS at an inappropriate speed can lead to misjudgments about its competence. Additionally, we explore the impact of computational delays, common in real-world deployments, on driving performance. We present a novel approach to counteract the effects of delays by adjusting the target labels in the training data, demonstrating improved resilience in models to handle computational delays effectively. This method, crucially, addresses the effects of delays rather than their causes and complements traditional delay minimization strategies. These insights are valuable for developing robust autonomous driving systems capable of adapting to varying speeds and delays in real-world scenarios.

## 1. Introduction

The fully end-to-end driving systems convert raw sensory inputs directly into actionable driving commands via a single neural network model [1,2,3,4]. Imitation learning, especially behavioral cloning, has been the dominant training paradigm for training such models [4,5,6,7] in the past ten years. The camera feed is often the only input [8,9,10,11,12,13]. As output, the model attempts to predict human-like steering angle, throttle, and brake values, i.e., “what the human expert would do in the given situation” [8,13]. These actionable “end” commands can be passed into the drive-by-wire system of the vehicle. Alternatively, end-to-mid approaches would output a desired trajectory [10], cost map [14,15], or other not directly actionable output representations [16], which need to be further processed by subsequent modules. In the following, camera-based fully end-to-end driving is employed, while some of the problems raised and solutions proposed generalize to a wider variety of approaches.

Over recent years, different neural network architectures have been employed for mapping inputs to outputs. Convolutional neural networks (CNNs) can efficiently process single-frame inputs [8,17]. To deal with multiple input timesteps, the images can be stacked and processed by a CNN [14,18] or can be encoded by a CNN and then analyzed in sequence by a recurrent part of the network [10,19,20]. Recently, visual transformers have emerged as a powerful tool for merging multiple input sources [7,21,22]. Using auxiliary tasks adds learning signals while also improving interpretability [6,14]. In the following, only CNN-based approaches are utilized due to their relatively low computing and data needs.

As a major limitation of behavioral cloning, the learned task of transforming randomly mini-batched images to actionable commands is not identical to the actual task of driving [6,23,24]. The task of predicting on a static dataset (open-loop testing, off-policy testing) and the task of controlling a vehicle in real time (closed-loop testing, on-policy testing) differ in the following aspects:The type of mistakes that matter differs. Driving is a sequential decision-making process. Small temporally correlated errors can make the vehicle drift off the safe trajectory little by little [25]. Behavioral cloning with usual loss functions and random minibatch sampling penalizes such consistent but weak biases minimally. Such errors look harmless in off-policy testing. Furthermore, the long-term negative effects of actions are not evaluated in off-policy testing [24].Incoming data differ. During deployment, the learned policy defines the distribution of situations the vehicle ends up in. This distribution may significantly differ from the training and test set situations resulting from expert driving. Behavioral cloning models trained on human driving may fail to generalize to the distribution caused by self-driving. This well-known effect is often called the distribution shift [1,24,26].Delays differ. Delays play no role in the predictive behavioral cloning task optimized when creating a model. The loss does not increase if the computation is slow. When deployed in the real world, delays exist between the moment an observation is captured by the camera and the moment the actuators receive the computed response to this observation. This has been discussed for model predictive control [27,28] but not for end-to-end approaches. Due to delays, optimal decisions may become outdated by the time they are applied.Decision frequency differs. Prediction frequency does not influence the loss in the behavioral cloning task the models are optimized to perform. The loss value does not increase if the computation is slow. However, during driving, infrequent decisions may overshoot their intended effect, resulting in an oscillation around the actual desired path. Furthermore, at low frequencies, the model can be simply too late to react to situations arising between decision ticks.

In our previous work [29], we noticed that the driving speed differed between the data collection drives (including data for off-policy evaluation) and deployment (on-policy testing). In particular, we deployed steering-only models on a real vehicle at speeds 0.5 to 0.8 times the speed the human used in the given GPS location. Similarly, ref. [11] used speeds only up to 10 km/h when deploying models. Indeed, a lower speed is safer for the safety driver and is, in many ways, a smart approach. However, the present work aims to demonstrate that deploying at unusually low speeds can cause an out-of-distribution problem for the steering models.

Intuitively, the on-policy sequential task of driving fast is a different task from driving slow, due to the physics of the world. At higher speeds, the vehicle slips more in turns and the forces counteracting actuation are higher, increasing actuation delays. Beyond physics, the computational delays result in decisions being increasingly late and sparse in terms of space as speed increases. These are intuitive and known effects that one can attempt to reduce by minimizing delays and optimizing hardware. Importantly, this work reveals that also the behavioral cloning task itself becomes different at different speeds. During data collection, the expert driver (human, or another model) must account for the chosen speed profile and the steering commands, i.e., the labels of the task, to differ. In particular, faster driving necessitates preemptive actions to counter inertia, delays in transmitting commands, actuator delays, and computational delays (within the brain if the data were collected by a human), while slow driving does not. To illustrate, while the camera image at a given location (e.g., 2 m before a turn) remains very similar at different speeds, the labels (the output a model should predict) differ. In the following, this is referred to as the task shift, a shift in the predictive function from inputs to outputs the model attempts to learn; in particular, a shift in the correct outputs for a given input. The severity of task shift between speed levels increases linearly with delays and with speed difference as explained in Figure 1.

Human drivers also turn preemptively, so this “task difference” is captured in the recorded data. Even at zero computational delays during deployment, models trained on fast data will be less adequate for driving slowly (and vice versa). There exists a difference in the image–label pairs defining the task for behavioral cloning, originating in the physics of the world and computational delays and decision frequencies of the data collector (e.g., human reaction time and decision frequency). As a consequence, testing an end-to-end autonomous driving system (ADS) under a novel speed profile will always pose a generalization challenge for the model.

Beyond steering models learned by behavioral cloning, the relevance of the problem of speed-induced task shift depends on the model type and how the self-driving task is formulated. Models relying on multiple frames or receiving speed as input will face input out-of-distribution challenges if deployed at a novel speed. Models controlling speed jointly with steering can learn to output adequate pairs for a given input if (1) exposed to them during training, and (2) the output speed is not artificially clipped in post-processing (for safety reasons, to ensure driving slow). Clipping results in slow speed being matched with a steering value intended for a higher velocity. Producing full joint probability distribution or energy landscape over speed and steering values would allow sampling pairs in a desired speed range without the need to clip the speed. In many end-to-mid approaches, the imitation learning task is very similar at different speeds (e.g., predicting trajectory and cost map) because counteracting the physics is left for the consecutive control module. However, at deployment, the predicted mid-level outputs will become outdated by the time they are produced and some correction of ego-location is needed.

In general, driving at different speeds always constitutes a different task for the ADS, if not due to computational delays, then due to slip and actuator delays (higher centrifugal force acting against the actuation). The presence of computational delays amplifies this difference during deployment, e.g., on-policy testing. Making decisions late in terms of time does not necessarily lead to crashes (e.g., at speed 0), but being late in terms of space (location on the road) does. Delay × speed, i.e., the spatial belatedness, is an important characteristic of the deployment task the vehicle is performing. Driving fast in the presence of minimal delay is surprisingly similar to driving slow in the presence of significant computational delay, as in both cases, decisions need to be made early in terms of location and hence in terms of camera inputs. Consequently, there are two widely applied ways to counteract spatial belatedness—driving slowly and minimizing delays.

Here, a third option is proposed—conditioning behavioral cloning models to match observation at time *T* to the label recorded at time T+δT, where δT is the expected computation time during deployment. This way, the produced command is relevant at the moment it actually gets applied. Technically, the training set labels must be matched with an earlier frame, so we name this approach label-shifting.

To safely demonstrate the effects of speed, computational delays, and the proposed label-shifting countermeasure, this work utilized 1:10 scale Donkey Car [30,31] mini-cars equipped with Raspberry Pi 4b and a frontal camera. The models were trained fully end-to-end with behavioral cloning and controlled only the steering of the vehicle. The training and deployment procedure is very similar to what was performed on the real-sized car in our previous work [29]. We believe the lessons learned are transferable to the domain of real-sized cars.

The main contributions of the present work are given as follows:1.We demonstrate that the performance of good driving pipelines may fall apart if deployed at a speed the system was not exposed to during training. The underlying reasons and the implications for on-policy testing are explained. To our knowledge, the effect of deployment speed has previously not been discussed in the end-to-end literature.2.We illustrate, via real-world testing, how the performance of good driving models suffers due to computational delays. The presented results demonstrate that label-shifting the training data allows to easily alleviate the problem, reducing the effect of delays. Incorporating delay anticipation into end-to-end models has not been attempted before.

## 2. Materials and Methods

In this work, Donkey Car S1 1:10 scale mini-cars were employed to study the effect of speed and the effect of computational delays. The practical part of the work was conducted as two master theses, independently (the two theses can be found at https://comserv.cs.ut.ee/ati_thesis/datasheet.php?id=74970&language=en (accessed on 1 December 2023) and https://comserv.cs.ut.ee/ati_thesis/datasheet.php?id=75358&language=en (accessed on 1 December 2023). In this section, first, the overall setup and hypotheses of the two sub-studies of this work are defined. Thereafter, descriptions are provided for the hardware used, for the methods of data collection and organization into datasets, for the used model architectures, and for the performance evaluation methods.

### 2.1. Experimental Design

#### 2.1.1. Study on the Effect of Speed

To demonstrate the effect of speed on model performance, behavioral cloning models [32] predicting the steering angle on data collected at a certain speed (low, high) were created, and their generalization ability to another, novel speed was evaluated. The low speed was chosen as the lowest possible speed achievable with the vehicle (insufficient torque at lower throttle values). The high speed was chosen to be the maximal comfortable speed for the human data collector. The resulting lap times are perceivably and statistically clearly different. Single-frame models considering only the latest frame and multi-frame models considering the past 3 frames were trained, the architectures of which are given below. We hypothesized that novel speed causes a performance drop in the ability to predict the labels in the off-policy behavioral cloning task, as well as in the on-policy driving task of driving on the track.

Furthermore, we hypothesized that multi-frame inputs become out-of-distribution (OOD) if the data originate from a novel speed. The input images are increasingly dissimilar from each other as speed grows, posing a generalization challenge for models that have developed visio-temporal features assuming a specific speed. In particular, we hypothesized that the out-of-distribution effect of novel speed inputs results in measurably different activation patterns inside the networks as shown for OOD inputs in other domains [33,34].

#### 2.1.2. Study on Counteracting the Effect of Delays via Label-Shifting

In the second study, the effect of computational delays during deployment was quantified and counteracted. The dataset was collected at high speeds to amplify the effect of delays and render the results more evident. This collected dataset was transformed into a variety of training sets by shifting the steering values (labels) by one or multiple positions back or forward in time, matching labels with previous or subsequent frames. This way, models that predict optimal commands for the past, present, or future were created. These models were deployed on the track in the presence of different amounts of computational delays, and their driving performance was measured.

In particular, the existing computational delay was increased artificially by inserting a waiting time (25 ms, 50 ms, 75 ms, or 100 ms) after neural network computing was finished and before sending out the command (via time.sleep() function). The longer compute delay imitates using a larger network, weaker compute environment, or more processes sharing the compute resources. We hypothesized that by default, a model’s ability to drive deteriorates quickly as delays increase. Additionally, we hypothesized that models predicting future commands can perform better in the presence of increasing delays, as they implicitly take into consideration their presence.

### 2.2. Hardware Setup and Data Collection Procedure

This work was performed using Donkey Car open-source software version 4.3.22 [30] deployed on the 1:10 scale Donkey Car S1 platform (https://www.robocarstore.com/products/donkey-car-starter-kit, accessed on 15 January 2022), equipped with MM1 control board, Raspberry Pi 4b, Raspberry Pi wide-angle camera (160°), and two servomotors (steering, throttle). The turning diameter was 140 cm, and the maximum speed was several meters per second. The vehicle, referred to as the mini-car from here on, was deployed on a track 60–80 cm wide and 17 m long (Figure 2). The data were collected, and deployment was performed always under the same light conditions, either cloudy afternoon or evening with artificial lighting.

During data collection, the steering was controlled by the researcher using a Logitech F710 gamepad or by a competent self-driving model running on board the mini-car (explained below). The messaging delay from the gamepad to Raspberry Pi is not perceivable. However, actuator delays become perceivable when driving fast, likely due to inertia and the friction of the wheels with the ground. Actuator delays and the reaction time (approximately 250 ms for humans [35]) of the driver result in different driving styles when driving fast and slow.

The throttle can be fixed at a constant value by the human operator. The speed resulting from a constant throttle value depends on the battery level and the heating up of the servomotors. The mini-car cannot measure and maintain constant speed. In this study, driving speed was defined via actual achieved lap times, not via the throttle value. During data collection and on-policy testing, the operator steadily increased the throttle value to achieve stable lap times as the car heated and batteries drained.

In all conducted driving experiments, all computation happened on board, in the Raspberry Pi 4b device. This holds for both on-policy testing and data collection performed with the help of a competent “teacher” model. The duration of different computations, including neural network computing time, was measured by a built-in function of the Donkey Car software v 4.3.22. Model training took place on CPUs in laptops or in Google Colab with GPU access and took up to a few hours per model.

#### Quality of Driving Data

During data collection, the authors found it difficult to give intermediate-valued commands with the Logitech F710 gamepad’s stick button, especially at higher speeds where the actions are rushed. The need for steeper turning to counter inertia was exaggerated by low reaction time and precision of movement. This risked creating an artificial difference between slow and fast driving data, caused by the operator’s ability and not inherent to the tasks.

To bypass this issue, a teacher agent capable of driving at various speeds was utilized in the part of this study comparing the tasks of driving fast and driving slow. The teacher model used the same architecture and training setup as the single-frame models defined below but had been exposed to a variety of speeds during training. The same teacher agent collected training datasets at two different (slow and fast, defined below) speeds.

### 2.3. Data Preparation

The cleaned (infractions removed, speed in the designated range) datasets were prepared to demonstrate the two effects.

Study of speed. In the slow speed dataset (19,250 frames), the 17 m lap was, on average, completed in 24.25 ± 1.9 s, i.e., 0.7 m/s speed. The fast dataset (20,304 frames) corresponded to an average 14.85 ± 0.8 s lap time, i.e., 1.1 m/s speed. Both these sets were collected by the teacher-agent driving in the evening time with artificial light. From these two sets of recordings, single and multi-frame datasets were created. In the latter, each data point consisted of three frames matched with the steering command recorded simultaneously with the third frame.Five-fold cross-validation was performed by dividing the data into 5 blocks along the time dimension. In off-policy evaluations, the average validation results across the five folds are reported. For multi-frame models, the data were split into several periods along the time axis, and a continuous 1/5 of each period was assigned to each of the 5 folds. For both model types, new models were trained on the entirety of the given-speed dataset for on-policy evaluations to make maximal use of the data and achieve the best possible performance.Study of counteracting the effect of delays by label-shifting. All data in this study were recorded by a very proficient human driver at an average speed of 8.32 ± 0.41 s per lap. Data were collected in the afternoon with no direct sunlight or shadows on the track. Datasets matching camera frames with commands recorded up to 100 ms before and up to 200 ms after the frame capture were created. In total, there are seven datasets with the labels shifted by −100 ms, −50 ms, 0 ms, 50 ms, 100 ms, 150 ms, and 200 ms (due to recording at 20 Hz, shifting by a position corresponds to 50 ms). Each dataset was divided into training and validation sets with a random 80/20 percent split (46,620 and 11,655 frames, respectively). The validation set was only used for early stopping.

### 2.4. Architectures and Training

In this work, relatively simple types of neural networks were chosen for use. Firstly, the computation must run in real-time on a Raspberry 4b device, restricting us to low image resolution and limited network depth. Advanced approaches like visual transformers have not been validated to perform under these restrictions, while the chosen architectures have been used by the Donkey Car community and are known to perform sufficiently well on similar hardware setups. Secondly, more complicated network types usually require more training data to converge. Practically, the simplest model types could perform the task and were sufficient for this study.

For investigating the effect of speed of model performance, four types of models were trained:1.Single-frame CNN architecture, trained on fast data.2.Single-frame CNN architecture, trained on slow data.3.Multi-frame CNN architecture, trained on fast data.4.Multi-frame CNN architecture, trained on slow data.

The study on label-shifting to correct the effect of delays used only the single-frame architecture. The architectures used are summarized in Table 1 and Table 2.

The default training options in the Donkey Car repository were used. The mean squared error (MSE) loss function and Adam optimizer with weight decay with default parameters were used. Early stopping was evoked if no improvement in the validation set was achieved in 5 consecutive epochs, with the maximum epoch count fixed to 100.

### 2.5. Evaluation Metrics in the Study of Speed

The off-policy metric used in the study of speed was the validation set mean absolute error (MAE) as averaged over the 5 folds of cross-validation. On-policy behavior was observed when deploying the models on the vehicle at a fixed low or high speed (the same two speed ranges as in the training data). The main on-policy metric was the number of infractions, i.e., collisions with walls, during 10 laps.

#### Measuring the Out-of-Distribution Effect

The following analysis intends to demonstrate that for multi-frame models, novel-speed validation data cause activation patterns in the network to become more distinct from the patterns generated by the training data than same-speed validation data. Prior works have shown that out-of-distribution inputs cause detectably different activation patterns (i.e., embeddings) in the hidden layers of a network [33,34].

To this end, for every multi-frame model in 5-fold cross-validation, for both speeds, the following hold:1.Using training data, the final embedding layer neuron activations in three possible locations on the computational graph are computed: (a) after the matrix multiplication, (b) after batch normalization (BN), and (c) after both BN and ReLU activation. For each possible extraction location, the analysis is run separately. These activation vectors are called the reference activations.2.Similarly, neural activations on the validation set data points of the same speed dataset are computed. The resulting activations are referred to as same-speed activations.3.Every validation sample is described by a measure of distance to the reference set, defined as the average distance to the 5 nearest reference set activation vectors. Euclidean and cosine distances are employed as the proximity measures, and a separate analysis is performed for each (Ref. [33] proposed to use Mahalanobis distance, but our experience shows a competitive performance across different datasets with these simpler metrics).4.The activation patterns for the entirety of the other-speed dataset are computed. These activation vectors are called the novel speed activations. The distances of these activation patterns to the reference set according to the same metric are computed.5.Approximately, the further the activation patterns are from the training patterns, the further out-of-distribution the data point is judged to be for the given model [33]. By setting a threshold on this distance, one can attempt to separate the same speed and novel speed activations. The assumption was that novel speed activations are more different and mostly fall above the set distance threshold. The AUROC of such a classifier is computed and presented as the main separability metric.

These steps are also presented as an algorithm in the Appendix A.

In the Results, the average Euclidean and cosine distances to the reference set for the same and novel speed validation data are reported, averaged over the 3 possible extraction points. Averaging over the extraction locations is performed because there is no a priori knowledge of which extraction location to choose. Additionally, for all possible combinations of model, metric, and extraction location, the AUROC metric is computed, quantifying if it is possible to separate activation patterns emerging in response to novel speed data from those resulting from the same speed data.

We acknowledge that despite conscious efforts to guarantee similar conditions, the lighting might have slightly changed between fast and slow data collection. Also, limited motion blur can be seen in the fast-speed data. These two sources can cause additional input distribution shift between the collected slow and fast-driving data, beyond the frame-difference increase. To eliminate these other sources of input change, synthetically sped-up validation data were generated by skipping frames in the validation recordings. This analysis was performed only for slow-speed data; methods to generate artificially slow data based on fast recordings are more complicated and do not guarantee perfectly in-distribution individual frames. For every slow-speed validation set image triplet, i.e., frames from timesteps (t, t + 1, t + 2), a matching sped-up validation triplet from timesteps (t, t + 2, t + 4) was constructed. The network activations and the distances to the reference set were computed (as explained above) for the triplets in these two validation sets. The resulting distance values in the two sets are paired and not independent. Hence, instead of measuring AUROC, a one-sided Wilcoxon rank sum test was applied to compare the two lists of distances. The procedure of generating artificially fast data and comparing the resulting distances was performed for every model (i.e., every validation set) in the 5-fold cross-validation, for all different distance metrics and activation vector extraction points.

### 2.6. Evaluation Metrics in the Study of Delays

As discussed earlier, driving slowly can compensate for higher compute delays, as the root cause of failure is spatially late decisions. It was also concluded that driving at different speeds is a different task, as is driving in the presence of different delays. Consequently, when studying the deterioration of performance as delay grows, the other factor, the speed, must be fixed.

As the first evaluation, it is determined which delay and label-shifting combinations allow the vehicle to drive at the training set speed, i.e., perform the original task in terms of speed. Here, driving at the designated speed is defined as not slower than the training set mean speed + 2 standard deviations (8.33+2×0.41 s).

As delays and speed combine multiplicatively to cause spatial belatedness and failure, a model capable of driving faster must have counteracted the effect of delays more effectively. The highest possible safe driving speed is therefore also a revealing metric. The highest possible speed is determined by deploying the model–delay pair and increasing the speed gradually until the vehicle starts to crash regularly. The speed is then readjusted to just a fraction slower, and the vehicle attempts around 25 laps at this highest speed. The average lap time over these laps is reported for each model–delay combination.

### 2.7. Code and Data Availability

The code and the measurements (absolute error values for off-policy evaluation, infraction counts) underlying the results of the study on the effect of speed can be found at https://github.com/kristjanr/dat-sci-master-thesis/tree/master (accessed on 1 December 2023). The code and the measurements (lap times) underlying the results for the study of label-shifting are available at https://github.com/ilmaruduste/donkey-car-implemented-lag (accessed on 1 December 2023).

## 3. Results

### 3.1. Changing Speed Causes a Shift in The Task

Four types of models (slow/fast training data; single/multi-frame architectures) were evaluated using on-policy and off-policy measures. All models were evaluated in the conditions of slow and fast driving, i.e., also in the speed condition not trained for.

Off-policy evaluation was performed via 5-fold cross-validation for same-speed data and on the entire dataset for novel-speed data. The results are presented in Table 3. In short, on average, the models perform better at speeds they are trained at. When transferring models trained on fast speeds to slower data, the decrease in performance is less pronounced. This is because all the trained models tend to underestimate steering values, and this, by chance, aligns well with the slower driving, requiring less extreme steering.

For on-policy evaluation, the models were deployed on the Donkey Car S1 hardware. The same track where the data were collected was used, and the constant throttle was tuned to achieve lap times matching the slow and fast data. The number of infractions (collisions) over 10 laps was counted. The results are presented in Table 4 and demonstrate that all models perform remarkably better at the speed they are trained. In particular, fast-data models deployed slowly tend to cut into the inside corner of turns, while slow-data models deployed fast turn too late and end up at the outside wall (see Figure 3).

In conclusion, despite slow driving being perceived as an easier task, across two different architectures (single and multi-frame), models trained to drive fast cannot reliably perform it. Failing to perform a simpler task supports the theoretical discussion about the existence of a task shift between slow and fast driving, presented in the Introduction.

#### Additional Cause: Multi-Frame Inputs Become Out-of-Distribution

Here, the fact that slow-driving input data are out-of-distribution (OOD) for the models trained on fast data and vice versa will be demonstrated. For network activations resulting from same-speed and novel-speed validation sets, mean distances to the reference set activations (i.e., training set activations), averaged over models and the three possible extraction locations, are presented in Table 5. The results suggest that novel speed consistently brings about more distinct, i.e., more out-of-distribution, activation patterns in the used networks. Fast data are especially problematic for slow-data-trained models.

It is possible to try to separate known speed (in-distribution) and novel speed (OOD) data points based on this distance by setting a threshold above which all data points are classified as OOD. For all combinations of the model, metric, and activation extraction point, the AUROC values remained between 0.718 and 0.932 and showed a non-trivial separability. In other words, the activation patterns caused by novel-speed data were always detectably more different from training patterns.

Additionally, artificial “faster” data were synthesized based on the five slow data validation sets in cross-validation via skipping frames when forming input frame triplets. This way, the individual images and the underlying trajectories are identical for the slow and artificial-fast data. For every original input triplet, there is also a synthetically fast triplet, so the data are paired. For all model, extraction layer, and metric combinations, the synthesized-fast data resulted in, on average, more distant activation patterns from the reference activations than the same-speed validation set (one-sided Wilcoxon rank sum test, *p*-value $<10^{-20}$ for all cases).

In conclusion, novel speed image triplets caused networks to activate in patterns that are different from the patterns seen during training. This likely contributes to the lower performance at novel speeds, as the model is not optimized to perform well in this novel region of the activation space. Both this effect and the task shift contribute to reduced off-policy and on-policy performance at novel speed.

### 3.2. The Effect of Delays Can Be Counteracted by Label-Shifting

A set of single-frame steering models was trained, each learning to match training dataset frames with a different set of labels. These labels were obtained by shifting the rows of the labels file to match the frame at *T* with the label at T+shift. Shifting by one row corresponded to a 50 ms shift in time due to 20 Hz recording. The on-policy performance of these models was measured in the presence of different amounts of computational delay.

Which models under which delay conditions could perform the original task, i.e., completing laps at a speed similar to the training data, was evaluated. Table 6 presents the fastest lap times one could reliably obtain with each model–delay combination. Results falling into the training set mean lap time + 2 standard deviations (8.32 ± 0.82), i.e., our definition of the original task, are marked in bold. The baseline model with no label-shifting could perform the task at 24 ms delay (the actual compute time) and 49 ms delay but not at delays of 74 ms and above. Hence, a system with similar driving ability but three times slower (e.g., larger network, larger input, and unnecessary image pre-processing) would fail to drive safely at the training set speed.

Models trained to predict future commands demonstrated an increased resilience to computational delays, except the model trained with 200 ms-shifted labels (discussed below). For example, the model trained to guess human actions 50 ms after the frame capture completed the task in the computational delay range of 24–74 ms (Table 1). The model predicting commands bound to occur 100 ms after frame capture completed the task with delays of 49–74 ms. In the experimental design used here, the delay also reduced decision frequency, making completing the task harder even if all outputs were adequate at the time the computation ended. The model trained with a 200 ms label shift would yield timely predictions for a delay setup resulting in 5 Hz decision-making, which is not sufficient and causes failure by itself. Additionally, the imitation learning task may become challenging, as the model attempts to predict human actions further in time from the frame capture. It is, therefore, somewhat understandable that models could rather perform at delays slightly lower than the label-shift.

We can already conclude that label-shifting successfully counteracted the effects of delays, allowing models to drive at the training speed in the presence of higher delays. This would allow slow-to-compute pipelines and low-compute-power environments to complete the task where the behavioral cloning approach would otherwise fail. This benefit costs nothing additional in terms of computing resources at deployment time.

Furthermore, Table 6 also shows the maximal speed each model was able to drive at under each delay condition. Speed and delays have multiplicative effects—driving slower counteracts the effect of delays. Consequently, a model capable of driving faster is more successful in dealing with the effect of delays.

The overall fastest driving was achieved with the model predicting labels 50 ms into the future in the condition of minimal possible computational delay (24 ms). If other delays (frame capture and transfer, actuator delays, etc.) were trivial, this model would deploy actions 26 milliseconds before a human would. Either this preemptive behavior or simply performing human actions more reliably than the human himself allowed this model to drive faster than the fastest laps in the training dataset.

Beyond this superhuman performance, the results in Table 6 show for all models the need to decrease speed when delay increases as expected from the theoretical discussion. Looking along the columns axis, label-shifted models tend to allow higher driving speed at equal delays, but only up to a certain time shift. A small label shift (in our case one frame, 50 ms) seems to never hurt performance. Predicting past labels makes the driving even more belated and requires lower driving speeds to compensate.

## 4. Discussion

### 4.1. Discussion of Results

Here, the impact of speed variations and computational delays on the performance of end-to-end steering models in autonomous driving systems (ADS) was explored, utilizing 1:10 scale mini-cars. Speed and delays combine multiplicatively to make decisions late in terms of space. Driving at different speed profiles or under different delay settings should be considered a different task that the model does not always generalize to. Practically, the results demonstrated that models trained at specific speeds exhibit decreased performance when tested at novel speeds. This poses a challenge for the real-life testing of end-to-end ADS because deploying models at human-like speeds increases risks [36].

Addressing model robustness, we suggest diversifying training data across speed ranges, though practical and legal constraints may limit this approach for real vehicles. Driving slowly may disturb traffic and is not legal in most locations. Despite diverse data, single-frame steering-only models would still fail to adapt steering to speed because one frame does not allow to distinguish the driving speed. Alternative strategies include integrating speed as an input to the model (this risks causing the inertia problem [6]) or employing architectures capable of deducing speed from multi-frame inputs. The joint prediction of speed and steering also allows for generating adequate steering–speed pairs.

Similarly to speed changes, also delay changes affect how early the turning must be decided to be initiated in a suitable location. The conducted experiments revealed that incorporating delay anticipation into the model training mitigated the adverse effects of computational lag at deployment and improved driving ability. Predicting future commands is cost-free and counteracts the negative effect of delays complementarily to other remedies such as decreasing compute time or vehicle speed. Compensating for deployment delays has been studied for control algorithms [27,28] but to our knowledge has not been discussed in the domain of end-to-end driving.

### 4.2. Limitations and Future Work

In the presented experiments, the effects of delays and reduced decision frequency were confounded. This problem formulation reflects the challenge we actually face in both mini-cars and real-sized vehicle (https://adl.cs.ut.ee/lab/vehicle, accessed 1 December 2023). The delay mainly comes from the neural network computation time, which cannot be parallelized easily. A future study could maintain decision frequency and only increase the delay to further investigate the usefulness of label-shifting. Other future topics include correcting also for actuator delays or estimating delay dynamically and correcting by a different margin at every timestep, similarly to what is performed for the controller in [27].

The outputs (trajectories, cost maps) of end-to-mid self-driving, the more popular design in the 2020s, are more stable across speeds and suffer less from speed-related task shifts. Counteracting actuator delays, inertia, and slip are handled mainly by the control module. However, all multi-frame models will be affected by the speed-induced input distribution shift, no matter if it is applied for end-to-end or end-to-mid driving or any perception subtask in the modular approach. Furthermore, counteracting deployment delays is equally relevant for end-to-mid solutions. Local plans and cost maps can become outdated or shifted in egocentric space before they get applied and could be corrected for any motion that happened while the inputs were processed.

All results of this work (speed-related task shift and input shift, correction of deployment delays) are equally relevant for real-sized self-driving vehicles employing a neural network-based approach. For example, deploying actions 50 ms late at 100 km/h speed means deploying them 1.2 m late. Such delay when entering a turn must be corrected by subsequent commands and adds lateral jerk, lowering passenger comfort and perceived safety [37,38]. In practical terms, acknowledging these effects can help understand why good predictive models underperform in certain on-policy tests. It also promotes choosing and designing approaches that are more robust to or actively correct for speed-related and delay-related task changes. We believe the presented discussions may aid in understanding the self-driving problem in general, providing novel angles such as thinking about spatial rather than temporal belatedness. 

## Figures and Tables

**Figure 1 sensors-24-01963-f001:**
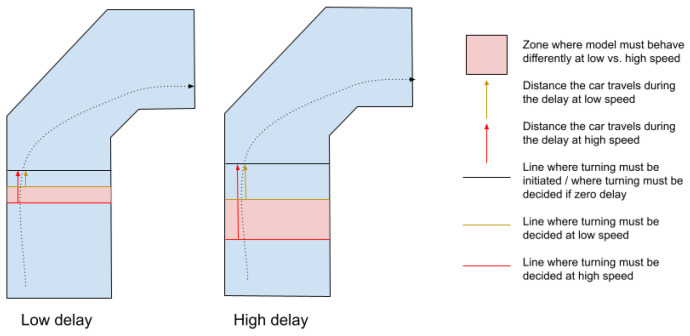
Preemptive actions. The command to turn must be decided preemptively based on the observation from delay · speed meters before the actual turning point. There exists an area, i.e., a set of observations (marked pink) along the road where the driver or driving model must behave differently at differing speeds. (Left to right) If delays grow, the difference between the tasks of slower and faster driving increases.

**Figure 2 sensors-24-01963-f002:**
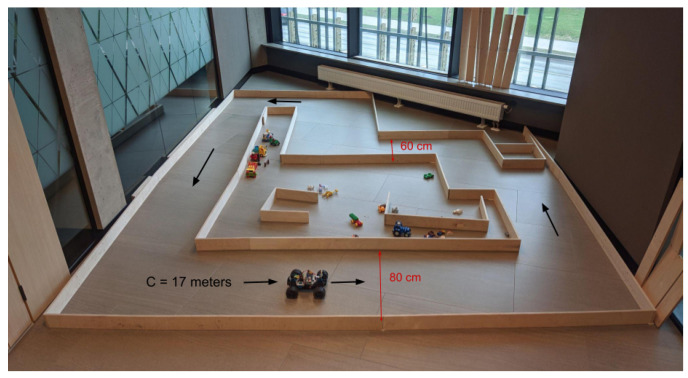
The track for data collection and on-policy evaluation. Driving was performed counterclockwise, resulting in five left turns and two right turns.

**Figure 3 sensors-24-01963-f003:**
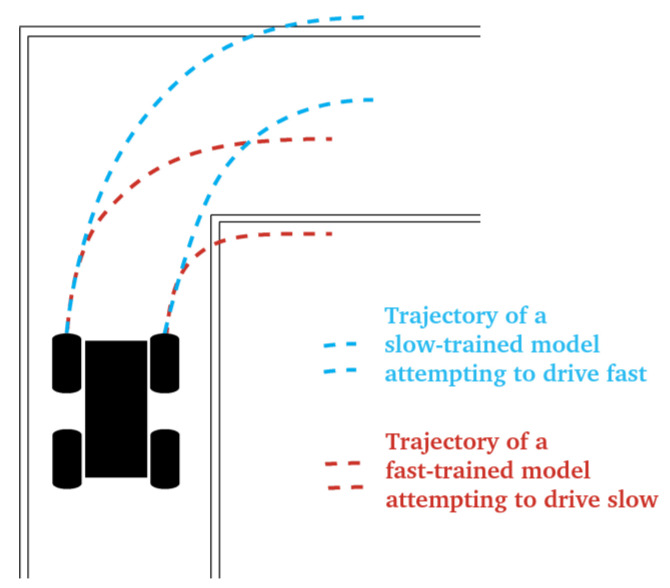
Common mistakes for solutions deployed at novel speeds. In blue: models trained on slow driving data fail to turn early and sharp enough when deployed at a fast speed. In red: models trained on fast driving data turn too early and sharply and hit the inside wall of the turn.

**Table 1 sensors-24-01963-t001:** Single-frame lateral-control CNN used in this work. This is the default architecture of the Donkey Car open-source software. Padding was always set to “valid”. The height is 60 pixels in the study of speed, and 120 pixels in the study of delays, as only the former uses cropping.

Layer	Hyperparameters	Dropout	Activation
Input	shape (height, 160, 3)	none	none
Conv2d	filters = 24, size = 5, stride = 2	0.2	ReLU
Conv2d	filters = 32, size = 5, stride = 2	0.2	ReLU
Conv2d	filters = 64, size = 5, stride = 2	0.2	ReLU
Conv2d	filters = 64, size = 3, stride = 1	0.2	ReLU
Conv2d	filters = 64, size = 3, stride = 1	0.2	ReLU
Flatten	-	-	-
Linear	nodes = 100	0.2	ReLU
Linear	nodes = 50	0.2	ReLU
Linear	nodes = 1	none	none

**Table 2 sensors-24-01963-t002:** Multi-frame lateral-control CNN used in the study of speed. Padding was set to “valid” in all cases.

Layer	Hyperparameters	Dropout	Activation
Input	size= (3, 60, 160, 3)	-	-
Conv3d	filters = 16, size = (3, 3, 3), stride = (1, 3, 3)	-	ReLU
MaxPool3D	pool_size = (1, 2, 2), stride = (1, 2, 2)	-	-
Conv3d	filters = 32, size = (1, 3, 3), stride = (1, 1, 1)	-	ReLU
MaxPool3D	pool_size = (1, 2, 2), stride = (1, 2, 2)	-	-
Conv3d	filters = 32, size = (1, 3, 3), stride = (1, 1, 1)	-	ReLU
MaxPool3D	pool_size = (1, 2, 2), stride = (1, 2, 2)	-	-
Flatten	-	-	-
Linear	nodes = 128, batch normalization	0.2	ReLU
Linear	nodes = 256, batch normalization	0.2	ReLU
Linear	nodes = 1	-	none

**Table 3 sensors-24-01963-t003:** Off-policy evaluation. Mean absolute error (MAE) metric for different architectures trained on different data, using validation data originating from fast and slow driving.

Model Type	Validation Data Speed	Mean Absolute Error
slow single frame	slow	0.0232
slow single frame	fast	0.0473
fast single frame	fast	0.0237
fast single frame	slow	0.0266
on average:	known	0.0235
	novel	0.0367
slow multi- frame	slow	0.0888
slow multi-frame	fast	0.1298
fast multi-frame	fast	0.0612
fast multi-frame	slow	0.0614
on average:	known	0.0754
	novel	0.0947

**Table 4 sensors-24-01963-t004:** On-policy evaluation. Infractions-per-ten-laps metric observed for different architectures trained on different data, deployed on the mini-car using two different speeds—fast and slow driving.

Model Type	Deployment Speed	Infractions
slow single frame	slow	0
slow single frame	fast	10
fast single frame	fast	2
fast single frame	slow	16
slow multi- frame	slow	0
slow multi-frame	fast	20
fast multi-frame	fast	8
fast multi-frame	slow	19

**Table 5 sensors-24-01963-t005:** Novel-speed inputs are out-of-distribution for multi-frame models. Mean five-nearest-neighbor distances to the training set activations are presented. In all cases, novel speed causes activations that are more dissimilar to the reference activations.

		Validation Data	
**Training Data**	**Metric**	**Same Speed**	**Novel Speed**
Slow	Euclidean	0.81	1.82
Slow	Cosine	0.004	0.019
Fast	Euclidean	0.81	1.21
Fast	Cosine	0.006	0.013

**Table 6 sensors-24-01963-t006:** Fastest lap times at different delay and label-shifting configurations. For each trial, the deployment speed is increased until the vehicle can no longer drive safely. Speed is adjusted to be just below that threshold, and 25 laps are attempted. The average lap time is reported based on recordings. Infinity signifies that the model cannot drive a collision-free lap at any speed. Results falling into the range of training set mean ± 2 standard deviations are marked in bold.

Compute Time	Used Label						
	**−100 ms**	**−50 ms**	**No Shift**	**50 ms**	**100 ms**	**150 ms**	**200 ms**
24 ms	10.3	**9.3**	**8.5**	**7.4**	*∞*	*∞*	*∞*
49 ms	11.9	9.7	**8.6**	**7.8**	**8**	*∞*	*∞*
74 ms	13.2	11.8	10.1	**9.1**	**8.5**	*∞*	*∞*
99 ms	16.2	13.8	11.5	10.5	9.4	**8.1**	*∞*
124 ms	17.5	13.7	12.5	11.3	10.6	**9.1**	*∞*

## Data Availability

The quantitative data about model performances presented in this work, e.g., infraction counts and error values on individual frames, can be found in the GitHub code repositories (cf. Code Availability). All data underlying the results tables can be found there. Underlying video recordings of experiments have not been stored and made publicly available due to their size. Further inquiries can be directed to the corresponding author.

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
