# Peer review of "The Effects of Speed and Delays on Test-Time Performance of End-to-End Self-Driving"

_sensors, 2024, doi:10.3390/s24061963_

Round 1

Reviewer 1 Report

Comments and Suggestions for Authors

The paper investigates the effects of speed variations and computational delays on the performance of end-to-end autonomous driving systems. I have couple of general comments:

1. Is there any previous research on "Delays differ" and "Decision frequency differs" in the introduction? If so, it is recommended to supplement the relevant literature in the introduction. All the papers you cite should be correct.

2. In "2. Materials and Methods", how to select the range of low speed and high speed of 1: 10 car model? What is the basis for the definition of low speed and high speed ?

3. The track data explained in Figure 2 has been mentioned in the paper, whether the relevant data of the test site can be marked in Figure 2.

4. Parts of 2.2.1, 2.5.1 and 3.1.1 appear separately and can be merged into the previous title without separate listing.

5. What is the "unit" of the data in Table 5?

6. Figure 3 lacks the meaning markers of different colored lines. It is recommended to mark on the right side of the figure as shown in Figure 1.

7. Is only 4.1 in "4. Discussion"? 4.1 Part of the content alone in the conclusion is not appropriate. The conclusion is not concise and clear enough.

Comments on the Quality of English Language

Minor editing of English language required.

Author Response

Honored reviewer,

we thank you for the time and effort put into the review. Multiple points in your opinion have been very useful in improving our manuscript. Please find below the point-by-point answers to the raised issues.

(original review text in bold)

The paper investigates the effects of speed variations and computational delays on the performance of end-to-end autonomous driving systems. I have couple of general comments:

  1. Is there any previous research on "Delays differ" and "Decision frequency differs" in the introduction? If so, it is recommended to supplement the relevant literature in the introduction. All the papers you cite should be correct.

We thank the reviewer for prompting this discussion. We have now added certain references from another field of study and commented on the lack of discussions in end-to-end literature on this topic. We have also changed the wording.

Indeed, in the domain of end-to-end driving, we could not find references for the intuitive observation that we do not care about compute frequency and compute speed in the off-policy evaluation metrics, but it plays a role during the on-policy evaluation. While a multitude of works discuss distribution shift (e.g. [Codevilla, Felipe, et al. "On offline evaluation of vision-based driving models."]), and several works mention that only in on-policy testing long-term consequences of actions play a role (e.g. [Haq, Fitash Ul, et al. "Can offline testing of deep neural networks replace their online testing? A case study of automated driving systems."] ), we could not find a work discussing the effect of compute time and delays. This is true even for works discussing the capabilities and differences between on-policy and off-policy testing (e.g. Haq et al.). 

However, the discussion on the effect of delays exists in the domain of model predictive control. In that domain, the researchers are aware of computational and actuator delays and look for ways to compensate for them [Liao, Yonglong, and Fucheng Liao. "Design of preview controller for linear continuous-time systems with input delay." ; Kalaria, Dvij, Qin Lin, and John M. Dolan. "Delay-aware robust control for safe autonomous driving and racing." ]. Notice that usually, the compute cycle is much faster in the control algorithms compared to an end-to-end neural network. Hence, for controllers the main source of computational delay is the “input delay” i.e. the time it takes for information to reach from sensors, via previous modules, to the controller. Researchers find it important to compensate for these delays, while end-to-end and end-to-mid communities do not correct for the potentially much larger delays in their pipelines.

The lack of this discussion about delays in the end-to-end self-driving community was surprising to us, especially given that neural networks can be notoriously compute-heavy. It is one of the core reasons for conducting this research. In our work, we provide this discussion on the effect of compute delays (Introduction) and go on to demonstrate the effects in the real world, as well as propose a potential remedy.

  1. In "2. Materials and Methods", how to select the range of low speed and high speed of 1: 10 car model? What is the basis for the definition of low speed and high speed?

We thank the reviewer for pointing out this disambiguity. We initially aimed for the slow and fast speeds to be 2x different. The exact values are unimportant, we simply needed a significant difference between the speeds. Due to its construction, the vehicle can not drive very slowly, it simply does not have enough torque at low throttle values. Hence the slow driving speed we can use was limited due to the Donkey Car platform. This speed allows driving at approximately 0.7 meters per second on average (7m/s or 25 km/h speed on the real cars’ scale).

The fast speed would have ideally been twice that, but was limited by the ability of the human operator collecting the data. As a result, the fast driving was on average 1.13 meters per second, or 40km/h on the real vehicle scale.

While the difference is not quite 2 fold as planned originally, the lap time distributions are not overlapping and the speeds are perceivably very different.

While this long discussion would be too verbose for the manuscript, we have now inserted the general idea into section 2.1.1.

  1. The track data explained in Figure 2 has been mentioned in the paper, whether the relevant data of the test site can be marked in Figure 2.

We agree that making figures self-explanatory is a good practice. We have now marked the driving direction, track width, and track length on the figure. 

  1. Parts of 2.2.1, 2.5.1 and 3.1.1 appear separately and can be merged into the previous title without separate listing.

We agree with the reviewer that the sections 2.5.1 and 3.1.1 repeat much of the content. As for section 2.2.1, we suppose the reviewer actually means 2.1.1 where the issue of multi-frame inputs becoming out-of-distribution for the models is first introduced.

By our logic, 2.5.1 defines an evaluation procedure and hence fits under the metrics/evaluation section. However, we admit that too much of the justification for this analysis as well as setting the hypothesis was contained in this Methods subsubsection. We have now corrected that and moved part of it to 2.1.1.

As the second point of concern, in the original manuscript the results section 3.1.1. indeed repated part of the content of Methods section 2.5.1. This was done in an attempt to make the results section more understandable on its own. However, we agree with the reviewer that part of that content actually belongs to the Methods.

We have now: 

  • moved the justification for the analysis and setting the hypothesis to section 2.1.1 (experimental design) while keeping the computing procedure in section 2.5.1.
  • reduced the amount of methods description in the Results section 3.1.1 and moved it to the Methods section 2.5.1. 
  1. What is the "unit" of the data in Table 5?

We appreciate the reviewer asking for clarification on this complicated point. The table 5 is a summarization of a complex analysis we performed to prove that not only changes in images but also changes in the inter-image difference can cause a generalization problem for the networks. This analysis compared neural activation patterns inside our networks in response to image sequences from known and novel speed. 

Neural activation values do not have a unit, hence also the Euclidean distance between activation value vectors has no unit. Similarly, the cosine value of the angle forming between two activation vectors has no unit. The reported values are average distances over many such activation pattern to activation pattern comparisons.

  1. Figure 3 lacks the meaning markers of different colored lines. It is recommended to mark on the right side of the figure as shown in Figure 1.

We thank the reviewer for noticing this shortcoming. We have now added a legend to the Figure 3.

  1. Is only 4.1 in "4. Discussion"? 4.1 Part of the content alone in the conclusion is not appropriate. The conclusion is not concise and clear enough.

We appreciate the suggestion by the reviewer. We agree that new content should not be introduced in the Discussion and that the conclusions should be concise about the results. We have now shortened the Conclusions section from approximately 1.5 pages to 1 page despite the addition of new points of discussion at the request of another reviewer. We have removed some reiteration of results while focusing on the main conclusions.

Reviewer 2 Report

Comments and Suggestions for Authors

·         The study presents an investigation into the impact of speed variations and computational delays on the performance of end-to-end autonomous driving systems (ADS).

·         Utilizing 1:10 scale mini-cars with limited computational resources, the study aims to demonstrate the significance of different driving speeds on the task of the driving model.

·         The findings suggest that varying driving speeds pose challenges to the generalization capabilities of systems trained at a singular speed profile.

·         Models trained for high speeds encounter difficulties when operating at slower speeds, and vice versa, indicating a need for improved adaptability.

·         It is crucial to highlight the potential misjudgments about the competence of an ADS when tested at inappropriate speeds.

·         The study also delves into the impact of computational delays, a common issue in real-world ADS deployments, on driving performance.

·         A novel approach is presented to mitigate the effects of delays by adjusting target labels in the training data, showcasing improved resilience in models.

·         This method focuses on addressing the effects of delays rather than their causes, providing a complementary strategy to traditional delay-minimization approaches.

·         The insights gained from this research are vital for the development of robust autonomous driving systems capable of adapting to varying speeds and delays in real-world scenarios.

·         It is important to emphasize the practical implications of the findings for the advancement of ADS technology.

·         Discussion on the experimental setup and methodology used to simulate speed variations and computational delays would provide clarity and context.

·         The study should elaborate on the specific challenges encountered by ADS when operating at different speeds.

·         Clear delineation of the criteria used to evaluate driving performance under various speed and delay conditions is necessary.

·         Addressing potential limitations in the experimental design, such as the scalability of findings to larger vehicles or real-world scenarios, is essential.

·         Recommendations for future research could include exploring alternative strategies for mitigating the impact of computational delays on ADS performance.

·         Collaboration with industry stakeholders could provide insights into real-world challenges faced by autonomous driving systems.

·         Ethical considerations, such as safety implications associated with speed variations and delays, should be addressed.

·         Providing practical guidelines for integrating the proposed approach into ADS development processes would enhance the study's utility.

·         The study should discuss the broader implications of its findings for the future of autonomous transportation and urban mobility.

·         Overall, the paper requires detailed revisions to clarify methodology, address potential limitations, and contextualize findings within the broader field of autonomous driving research.

Author Response

Honored reviewer,

We thank you for the insightful review multiple points of which have helped us improve our manuscript. Please find below the point-by-point answers to the raised issues.

(original review text in bold)

 The study presents an investigation into the impact of speed variations and computational delays on the performance of end-to-end autonomous driving systems (ADS). Utilizing 1:10 scale mini-cars with limited computational resources, the study aims to demonstrate the significance of different driving speeds on the task of the driving model. The findings suggest that varying driving speeds pose challenges to the generalization capabilities of systems trained at a singular speed profile. Models trained for high speeds encounter difficulties when operating at slower speeds, and vice versa, indicating a need for improved adaptability. It is crucial to highlight the potential misjudgments about the competence of an ADS when tested at inappropriate speeds.

       The study also delves into the impact of computational delays, a common issue in real-world ADS deployments, on driving performance.   A novel approach is presented to mitigate the effects of delays by adjusting target labels in the training data, showcasing improved resilience in models. This method focuses on addressing the effects of delays rather than their causes, providing a complementary strategy to traditional delay-minimization approaches. The insights gained from this research are vital for the development of robust autonomous driving systems capable of adapting to varying speeds and delays in real-world scenarios.

We appreciate the reviewer’s position that our research has provided vital insights into the field. This provided summary captures well the key aspects of our work.

The reviewer brought out the following points of improvement.

It is important to emphasize the practical implications of the findings for the advancement of ADS technology. 

Providing practical guidelines for integrating the proposed approach into ADS development processes would enhance the study's utility.

Addressing potential limitations in the experimental design, such as the scalability of findings to larger vehicles or real-world scenarios, is essential.

We group together the above three comments from the reviewer. We agree that the practical implications are important to point out to the readers. In our case, there are two main implications: 

  • One should test the ADS in conditions it was optimized to perform in. Seemingly easier tasks, e.g. slower driving, may not be easier for a model not optimized to perform them.
  • Once we acknowledge the existence of delays during deployment we can start inventing ways to combat the effect of those delays. Predicting what the human expert human would do 50 ms after the input was captured is not a significantly more complicated prediction task, but it allows compensating for the delays bound to happen. 

If the sum of all delays in a self-driving vehicle amounts to 100 ms, at 100km/h speed the decided actions would get applied 2.4 meters late. At the minimum, such spatial delays create the need to correct for them in the subsequent commands, causing more lateral movement and passenger discomfort. In modular pipelines, the correction for “input delay” is possible in the control module (input to the control module is delayed by computations and information transfer of previous modules). In end-to-end approach, the correction must happen inside the only module there is, the end-to-end network. We know of no work doing such correction in the end-to-end domain prior to our work.

We have now added to the Discussion a paragraph on these implications:

The work is equally relevant for real-sized self-driving vehicles employing a neural network-based approach. The autopilot must behave differently at different speeds, multi-frame inputs change distribution as speed changes, and delays occur during deployment. Acknowledging these effects can help understand why models underperform in certain on-policy tests. It also promotes choosing and designing approaches that are more robust to or actively correct for speed-related and delay-related task changes. 

Discussion on the experimental setup and methodology used to simulate speed variations and computational delays would provide clarity and context.

We appreciate the need for further clarity on these points. In the Methods section 2.1.1 we have now included an explanation of how fast and slow speeds were defined. Also, in section 2.1.2. we have now expanded the description of how additional compute delay was inserted into the deployment pipeline. 

The study should elaborate on the specific challenges encountered by ADS when operating at different speeds.

Indeed, driving at different speeds is a different task. In the Introduction we now state:

“Intuitively, the on-policy sequential task of driving fast is a different task from driving slow, due to the physics of the world. At higher speeds, the vehicle slips more in turns and the forces counteracting actuation are higher, increasing actuation delays. Beyond physics, the computational delays result in decisions being increasingly late and sparse in terms of space as speed increases.”

To our knowledge driving slow poses no particular physical challenges beyond simply being different from faster driving. This difference from the point of view of a model attempting to predict adequate commands is explained in Figure 1.

Clear delineation of the criteria used to evaluate driving performance under various speed and delay conditions is necessary.

We agree with the reviewer in this statement. We have described the evaluation criteria in the Methods sections 2.5, 2.5.1, and 2.6. For measuring the effects of novel speed on model performance we used very standard metrics - MAE for off-policy tests and infraction count for on-policy tests. Similarly, when studying the effect of delays we could have simply tested the models at training speed and count infractions. This would have shown in a very discrete manner if the model can perform the task. However, our maximal safe driving speed metric is more continuous and allows showing a clear relationship between maximal speed, delay amount, and the amount of delay correction. Hence, while using different metrics across studies hinders understanding, the choice of metrics here was necessary.

We have attempted to improve the wording in the metrics sections in the new version of the manuscript.

 Recommendations for future research could include exploring alternative strategies for mitigating the impact of computational delays on ADS performance.

We agree that our current approach is the simplest possible approach for mitigating the effect of delays. In the relevant literature about correcting delays in control algorithms, the amount of delay is predicted on the fly as it may depend on inputs and situation. Also, actuator delays vary depending on vehicle state and the amount of correction needed varies. Surely, in neural-network driven solutions similar adaptive correction and the correction of actuator delays would be feasible. We have added a similar discussion in the manuscript:

“ Other future topics include investigating and correcting for actuator delays or attempting to estimate delay amount dynamically and correct by a different margin at every timestep, similarly to what is done for the controller in []”

Ethical considerations, such as safety implications associated with speed variations and delays, should be addressed.

We thank the reviewer for pointing out this important topic. We have added a mention of passenger comfort and perceived safety. We have also changed the wording and provided a reference about safety testing.

The study should discuss the broader implications of its findings for the future of autonomous transportation and urban mobility.

We thank the reviewer for this suggestion. We believe the strongest influence could be in improving the competitiveness of end-to-end systems in on-policy testing by avoiding the pitfalls of deploying at the wrong speed or not considering delays. In an early stage of a project, an unexplained failure at deployment can discourage further interest in the topic and limit participation. We have witnessed this in our students, as well as our own lab towards different tested approaches.

Our manuscript now reads: “In practical terms, acknowledging these effects can help understand why good predictive models underperform in certain on-policy tests. It also promotes choosing and designing approaches that are more robust to or actively correct for speed-related and delay-related task changes. “

Overall, the paper requires detailed revisions to clarify methodology, address potential limitations, and contextualize findings within the broader field of autonomous driving research.

We believe we have taken significant steps towards these three directions. We remain awaiting further comments and suggestions.

Reviewer 3 Report

Comments and Suggestions for Authors

The proposed manuscript presents an investigation about the effects of speed variations and computational delays on the performance of end-to-end autonomous driving systems.

The paper requires many improvements. The main problem is that it should be reorganized and many parts should be described in much more detail.

The Introduction provides a fairly good overview of the problem, but other information with appropriate references should be also added (e.g., in relevant works how is the information provided by the camera processed and utilized).
The contributions are given at the end of this section, but the novelties should be pointed out more clearly.

Section 2 and section 3 present the experimental setup and the obtained results, respectively. These parts should be reorganized since it is not logical in its current form, and many parts should be explained in much more detail. Some examples: in 2.1.2., neural network computation time is mentioned, but it not presented until that point what kind of neural network is used in the study; recordings are mentioned before the measurement system is presented; etc.
Master theses are not real scientific publications so the relevant content of these two works should be explained in the paper.
"We did this for models considering only the latest frame and for multi-frame models considering the past 3 frames." - What was the reason for using only these setups? What kind of setups are used in relevant works?
"computational delay related to neural network computation is on average 24 ms" - How was this determined?
"Actuator delays and reaction time (approximately 250ms for humans)" - Based on what was this statement given? At least a reference should be added.
Why were these setups chosen for the frame numbers? Why did the multi-frame version provide lower performance than the single-frame version?
"We also synthesized artificial "faster" data by skipping every other frame in the slow data" - The results should be added to the paper and clearly referred to in this part.
Do environmental changes affect the performance? Cameras are sensitive to lighting conditions. Were these effects taken into account during the data acquisition?

Other comments:
-Do not use pronouns "we" and "our" in the text, use passive voice instead.
-Figure captions are too long. The descriptions should be in the text, not in the caption.
-The used performance metrics should be defined in detail.
-The reference list should be improved since it contains too many papers from arXiv.

Comments on the Quality of English Language

Only minor changes required.

Author Response

Honored reviewer,

we thank you for an insightful review that has certainly improved our manuscript in content and in style. In the following we go over the points raised one-by-one, hoping to provide clarity.

The proposed manuscript presents an investigation about the effects of speed variations and computational delays on the performance of end-to-end autonomous driving systems. The paper requires many improvements. The main problem is that it should be reorganized and many parts should be described in much more detail.

We appreciate the work put into reading and understanding our work. This review demonstrates the reviewer’s knowledge about the domain, mentioning aspects such as the effect of light conditions on performance. Many of the mentioned details were originally omitted in the manuscript to keep the text short. Many journals favor shorter works. We have now tried to expand certain sections and add details while keeping the text compact.

The Introduction provides a fairly good overview of the problem, but other information with appropriate references should be also added (e.g., in relevant works how is the information provided by the camera processed and utilized). The contributions are given at the end of this section, but the novelties should be pointed out more clearly.

We thank the reviewer for an overall positive evaluation of our problem statement. We have now clarified the novelty of our contributions. 

We also acknowledge that beyond stating what the inputs and outputs of a model can be (first paragraph of the Introduction), the possible model architectures should be mentioned. The diversity of architectures is large, especially as the border between end-to-end and end-to-mid approaches is foggy. In the new version of the manuscript, we listed a few architecture choices in the second paragraph of the Introduction just to clarify for the reader that there exists a diversity of architectures. Providing an exhaustive list would be out of the scope of this work.

Section 2 and section 3 present the experimental setup and the obtained results, respectively. These parts should be reorganized since it is not logical in its current form, and many parts should be explained in much more detail.

Some examples: in 2.1.2., neural network computation time is mentioned, but it not presented until that point what kind of neural network is used in the study; recordings are mentioned before the measurement system is presented; etc.

We agree with the reviewer that our Methods section has a non-standard structure. This is mainly due to the fact we combine two sub-studies in our manuscript. We believe the two studies are complementary and we wish to emphasize that speed and delays are the two sides of the same coin. 

On the suggestion of the reviewer we experimented with moving Methods subsections around. Indeed, the hardware and network architectures could be placed at the beginning of the methods section. However, we found that the existence of two types of data and two sets of metrics would be confusing without first explaining the goals and hypothesis of the two sub-studies.

We have decided to stick with the following strategy in explaining the work:

  • State the overall goals and hypothesis for the two sub-studies
  • Define the data and models
  • Define the metrics  

To accommodate the suggestions by the reviewer, we have added an explanation about the logic of the Methods section to the very beginning of the section, guiding the reader about the logic we employ. The manuscript now reads:
In this section, first, the overall setup and hypotheses of the two sub-studies of this work are defined. Thereafter descriptions are provided for the hardware used, for the methods of data collection and organization into datasets, for the used model architectures, and for the performance evaluation methods.

Furthermore, we have tried to move the maximum of technical details away from the experimental design description of the two sub-studies and only convey the overall idea and hypothesis. Details such as the recording frequency and label shifting amount are now included in the data preparation section. 

We also ask the reviewer to consider that we had already submitted responses to the other reviewers before we noticed this third review had been added. We were reluctant to make drastic changes to the manuscript such as reordering of the Methods which would make the already submitted responses invalid.

Master theses are not real scientific publications so the relevant content of these two works should be explained in the paper.

We thank the reviewer for this clarification. We have now mentioned the thesis links in the footnote without improperly referencing them. The source of this mistake was lack of experience, not ill intent.

"We did this for models considering only the latest frame and for multi-frame models considering the past 3 frames." - What was the reason for using only these setups? What kind of setups are used in relevant works?

We agree with the reviewer that architecture choices should be justified. We have now added a paragraph in the “Architectures and training” subsection explaining the need for small architectures due to restricted computing power. With another version of the Donkey Car hardware utilizing Jetson Nano as on-board computer, more powerful networks could have been applied.

It is plausible that a miniature version of visual transformers would be fast enough to compute on Raspberry Pi 4b. However, developing and validating a novel model architecture was not the goal of this work. We chose model types that the Donkey Car community has used and validated for Raspberry Pi 4b.

"computational delay related to neural network computation is on average 24 ms" - How was this determined?

We have now clarified the sentence about computational delay measurement in the manuscript, now clearly stating that this measurement is done by the Donkey Car software stack. The statistics of compute times of every part of the pipeline are displayed at the end of a deployment session (on program closing).

"Actuator delays and reaction time (approximately 250ms for humans)" - Based on what was this statement given? At least a reference should be added.

We have now added a reference to the human visual reaction time. 

Why were these setups chosen for the frame numbers? Why did the multi-frame version provide lower performance than the single-frame version?

We agree that the choice of architectures was not explained sufficiently in the original version of the manuscript. We have now justified the choice of architectures in the “Architectures and training” subsection of the updated manuscript. Essentially, we need to keep computing time and number of parameters low. First, because of restricted compute power at deployment. Second, larger models require more data.

We can only hypothesize why the multi-frame models performed worse. In on-policy testing, the compute time is somewhat slower, likely affecting the performance. However, also the off-policy metrics are weaker. We assume the 3x larger input size (3 images instead of 1) results in an overparametrization and overfitting of the model. 20 000 frames was likely not enough. 

"We also synthesized artificial "faster" data by skipping every other frame in the slow data" - The results should be added to the paper and clearly referred to in this part.

In this point, the reviewer has truly pushed us to improve and correct our work. Indeed, this analysis is a very straightforward way to see if the frame difference plays a role, independently of motion blur or some hard-to-visually notice difference between conditions during slow and fast data collection. 

When revisiting this analysis at the request of the reviewer we realized that AUROC is not a suitable metric for comparing distances from original and synthetically fast validation data, because the samples are paired and not independent. The two validation sets have frame triplets starting from the same frame at different speeds (e.g. frames 0, 1, and 2 originally, and  0,2,4 in the sped-up data).  The matching of pairs was done based on the first frame of the triplets, as an arbitrary choice. It could similarly have been done by the middle or last frames. 

Due to the pairedness of the data,  we have now applied a one-sided Wilcoxon rank sum test on the two sets of distance-to-reference-set measures. For both validation sets, the frame triplets were passed through the network and the resulting activation patterns were compared with training set activation patterns, the distances to the set of five most similar training-set activations were computed and averaged. Wilcoxon rank sum test was applied on these lists of distances, effectively comparing distances from triplets of the type (t, t+1, t+2) with distances of triplets of type (t, t+2, t+4). 

The results demonstrate an increased distance to reference set activations for the frame-skipped triplets compared to the original-speed triplets. 

We have described this analysis in the Methods and present the results in the Results sections.

Do environmental changes affect the performance? Cameras are sensitive to lighting conditions. Were these effects taken into account during the data acquisition?

We thank the reviewer for raising this point. Indeed, for camera-only driving, light conditions strongly influence the models' performance. We have witnessed this for indoor mini-car as well as outdoor real-sized car end-to-end models. Beyond the ability to generalize to new light conditions, certain conditions are inherently more tricky, e.g. sharp shadow edges seem to trick the convolutional networks into believing there is a boundary.
The reviewer correctly assumes we have considered this during data collection and on-policy evaluation. We have now added this clarification about selecting light conditions during data collection and evaluation.

Other comments:

-Do not use pronouns "we" and "our" in the text, use passive voice instead.

We appreciate the help and suggestions on scientific writing in English. We have now tried to limit the use of active voice, but could not remove it everywhere.

-Figure captions are too long. The descriptions should be in the text, not in the caption.

We suppose this comment is mainly about Figure 1. In our school of writing, we have been taught that people skim through papers and look at figures first, so figures should be understandable on their own. Hence our figure captions are rather complete. However, we agree that the caption of Figure 1 was excessive and have reduced it now. Also in other figure and table captions, sentences repeating the text were removed.

-The used performance metrics should be defined in detail.

We agree with this, surely. With the actions taken in response to this review and other comments from all reviewers, we hope the clarity has increased.

-The reference list should be improved since it contains too many papers from arXiv.

We thank the reviewer for paying attention to this fact. We have now discovered that certain of these sources have actually been published in peer-reviewed journals, but our references list was not updated about this. We have corrected those references.

Nevertheless, some sources are only published in arXiv. We would like to point out that the remaining six pre-prints have a significant amount of citations and are well-accepted and respected works in the field. The work done not in academic institutions but in private companies is often only uploaded to arXiv.

Remaining arXiv sources:

  •  "Deep anomaly detection with outlier exposure"   1405 citations as per Google Scholar
  • "End to end learning for self-driving cars"     4807 citations as per Google Scholar
  • "Chauffeurnet: Learning to drive by imitating the best and synthesizing the worst"     753 citation as per Google Scholar
  • "Autonomous driving with deep learning: A survey of state-of-art technologies"   85 citations as per Google Scholar
  • "End-to-end autonomous driving: Challenges and frontiers"   46 citations as per Google Scholar
  • "Learning Accurate, Comfortable and Human-like Driving"  30 citations as per Google Scholar

We hope you find these responses and the introduced changes to our manuscript helpful and sufficient. 

With best regards,
The authors

Round 2

Reviewer 2 Report

Comments and Suggestions for Authors

Accept.

Reviewer 3 Report

Comments and Suggestions for Authors

The authors significantly improved the manuscript based on the recommendations of the reviewers and answer the questions. So, I think the manuscript can be accepted in its current form.

Comments on the Quality of English Language

Only minor editing required.